# PAGET: Hierarchical Multi-Teacher Knowledge Distillation for Comprehensive Tumor Microenvironment Segmentation

**Daisuke Komura**[1]                                    KOMURA@M.U-TOKYO.AC.JP
[1] *Department of Preventive Medicine, Graduate School of Medicine, The University of Tokyo, Tokyo, Japan*

**Maki Takao**[2]                                        MAKI.T.ESPOIR@GMAIL.COM
[2] *Department of Obstetrics and Gynecology, Graduate School, Tokyo Medical and Dental University, Tokyo, Japan*

**Mieko Ochi**[1]                                        NOONECANPASS.AND@GMAIL.COM
**Takumi Onoyama**[3]                                    T-ONOYAMA@TOTTORI-U.AC.JP
[3] *Division of Gastroenterology and Nephrology, Department of Multidisciplinary Internal Medicine, School of Medicine, Faculty of Medicine, Tottori University, Tottori, Japan*

**Hiroto Katoh**[1]                                      HKAT-PRM@M.U-TOKYO.AC.JP
**Hiroyuki Abe**[4]                                      ABEH-PAT@H.U-TOKYO.AC.JP
[4] *Department of Pathology, Graduate School of Medicine, The University of Tokyo, Tokyo, Japan*

**Hiroyuki Sano**[5]                                     H.SANO@BIOMY-TECH.COM
[5] *Biomy Inc., Tokyo, Japan*

**Teppei Konishi**[5]                                    T.KONISHI@BIOMY-TECH.COM
**Toshio Kumasaka**[6]                                   TSKUMASAKA@GMAIL.COM
[6] *Department of Pathology, Japanese Red Cross Medical Center, Tokyo, Japan*

**Tomoyuki Yokose**[7]                                   KAMABOKO8TY@GMAIL.COM
[7] *Department of Pathology, Kanagawa Cancer Center, Kanagawa, Japan*

**Yohei Miyagi**[8]                                      MIYAGI.0E82R@KANAGAWA-PHO.JP
[8] *Kanagawa Cancer Center Research Institute, Kanagawa, Japan*

**Tetsuo Ushiku**[4]                                     USIKUT@GMAIL.COM

**Shumpei Ishikawa**[1,9]                                ISHUM-PRM@M.U-TOKYO.AC.JP
[9] *Division of Pathology, National Cancer Center Exploratory Oncology Research & Clinical Trial Center, Chiba, Japan*

**Editors:** Accepted for publication at MIDL 2026

## Abstract

Comprehensive characterization of the tumor microenvironment (TME) from H&E-stained histopathology images remains challenging due to the diversity of cellular components and limitations of current segmentation methods. We present PAGET (Pathological image segmentation via AGgrEgated Teachers), a multi-teacher knowledge distillation framework that enables simultaneous segmentation of 13 TME components from a single efficient model. Our key insight is that teacher predictions should be aggregated following the biological taxonomy of cell types—from tissue-level context through major cell categories to specific subtypes—rather than simple voting. By training specialized teachers on immunohistochemical restaining data and distilling their aggregated knowledge, the resulting student model not only matches but consistently outperforms the teacher ensemble on external datasets. We provide two complementary variants: PAGET-S for rapid semantic segmentation and PAGET-H for detailed panoptic segmentation. Extensive evaluation across three external datasets demonstrates robust generalization. Our implementation is available at https://github.com/dakomura/PAGET.

**Keywords:** Knowledge distillation, multi-teacher learning, tumor microenvironment, histopathology, semantic segmentation, panoptic segmentation

## 1. Introduction

The tumor microenvironment (TME) orchestrates cancer progression through complex interactions among epithelial cells, immune infiltrates, and stromal components (De Visser and Joyce, 2023; Binnewies et al., 2018). Quantifying these diverse cellular populations from routine H&E slides could transform large-scale biomarker discovery and clinical decision support. However, achieving comprehensive TME characterization with both biological fidelity and computational efficiency remains an open challenge.

Current deep learning approaches for histopathology segmentation face three interconnected limitations. First, existing methods typically identify only 3–6 cell types, insufficient for comprehensive TME analysis. Second, these methods rely on morphology-based annotations by pathologists, which can be inaccurate for cells with atypical morphology where even experts cannot make definitive identifications. Immunohistochemical (IHC) restaining techniques address this annotation challenge by enabling protein-based ground truth (Komura et al., 2023), but introduce a third limitation: separate models must be trained for each antibody-cell type pair, making comprehensive TME analysis computationally prohibitive for large-scale whole slide image (WSI) analysis.

Addressing these limitations requires annotation accuracy, comprehensive cellular coverage, and processing speed. We achieve all three through multi-teacher knowledge distillation—aggregating IHC-trained teachers into unified supervision and distilling their knowledge into a single efficient student. A key insight is that cellular classification follows an inherent biological hierarchy: leukocytes subdivide into lymphoid and myeloid lineages, then into specific subtypes (Murphy and Weaver, 2016; Diehl et al., 2016). Rather than flat classification that ignores these relationships, our framework aggregates teacher predictions following this taxonomy.

Our contributions include: (1) a hierarchical aggregation strategy that combines teacher predictions following biological taxonomy; (2) the first unified framework for 13-class TME segmentation from H&E slides, with semantic (PAGET-S) and panoptic (PAGET-H) variants; (3) demonstration that the student consistently outperforms the teacher ensemble on external datasets; and (4) extensive validation across three diverse datasets showing robust generalization.

## 2. Related Work

### 2.1. Histopathology Image Segmentation

Deep learning has revolutionized automated analysis of histopathology images. Hover-Net (Graham et al., 2019) achieve simultaneous nuclear segmentation and classification through multi-task learning, while HD-YOLO (Rong et al., 2023) applies object detection paradigms to cell identification. Cerberus (Graham et al., 2023) demonstrates that a single model can perform multiple segmentation tasks. However, these methods typically focus on limited cell type repertoires (3-7 classes) and single-tissue contexts, resulting in incomplete TME characterization.

A fundamental challenge is annotation quality. Morphology-based annotations by pathologists can be unreliable for cells with atypical appearance (Komura et al., 2023). IHC restaining addresses this by enabling protein-based ground truth, but requires training sep-

arate models per antibody, making comprehensive analysis computationally prohibitive. Our work bridges this gap by distilling multiple IHC-trained specialists into a single unified model.

## 2.2. Knowledge Distillation

Knowledge distillation (Hinton et al., 2015) transfers knowledge from complex teacher models to efficient student models and has been widely explored in computer vision. Beyond single-teacher settings, ensemble and multi-teacher distillation methods compress the predictions or features of several teachers into a single student( (Shen et al., 2019), (Yang et al., 2025), (Ye et al., 2024)). However, these approaches generally assume that all teachers solve the same task and share an identical label space, focusing on fusing complementary views of a single prediction problem.

Our setting differs fundamentally: teachers are specialized for distinct biological entities (e.g., epithelium vs. specific immune subtypes) with heterogeneous output spaces. We propose taxonomy-aware aggregation that respects the hierarchical relationships among cell types, a formulation unexplored in medical imaging to our knowledge.

## 3. Method

### 3.1. Problem Formulation

Given an H&E-stained histopathology image, our goal is to produce a segmentation map covering 13 TME components plus background. We achieve this through multi-teacher knowledge distillation, aggregating specialized teacher predictions into unified supervision for a single efficient student model.

### 3.2. Dataset Construction for Distillation

Our distillation dataset comprises 59,443 H&E images from tissue microarrays spanning 22 cancer types, originally collected for IHC restaining studies (Komura et al., 2023) . We apply the teacher ensemble (Section 3.3) to these images to generate pseudo-labels for 13 TME components. The 13 classes span two annotation levels: tissue-level labels (epithelium, stroma, smooth muscle) and nucleus-level labels (epithelial cells, fibroblasts, endothelial cells, red blood cells, lymphocytes, plasma cells, myeloid cells, eosinophils, neutrophils, and mitotic cells). The dataset contains 8.7 billion labeled tissue pixels and 15.4 million predicted nuclei, with detailed statistics in Table 2 (Appendix A).

### 3.3. Teacher Model Ensemble

Our teacher ensemble comprises specialized models operating at two scales (Figure 1). At the tissue level, SegPath models (Komura et al., 2023) trained on IHC-restaining data provide pixel-wise segmentation of epithelium, smooth muscle, endothelium, and red blood cells. At the nucleus level, additional SegPath models identify leukocyte nuclei (CD45+), while dedicated granulocyte models trained on MPO and ECP staining distinguish neutrophil and eosinophil nuclei. MIDOG++ (Aubreville et al., 2023) contributes mitotic figure detection. HoverNet (Graham et al., 2019) serves a dual role: it provides nucleus

instance masks that define spatial boundaries for aggregation, and contributes baseline 6-class nucleus classification (trained on PanNuke(Gamper et al., 2020)) that is refined by more specific teachers in our hierarchical aggregation. Architectural details for all models are provided in Appendix B.

### 3.4. Teacher Prediction Aggregation

Our aggregation strategy is motivated by the assumption that higher-level biological categories (e.g., tissue context or major cell lineages) are more robustly predicted from H&E images than fine-grained subtypes. This assumption naturally arises in settings where teacher models are trained on heterogeneous markers and operate at different semantic resolutions, as in our IHC-trained ensemble with partially overlapping label spaces. Hierarchical aggregation exploits this asymmetry by constraining subtype predictions using more reliable coarse-level outputs, which is not possible in flat aggregation.

Specifically, PAGET aggregates teacher predictions following a biological taxonomy of cell types (Figure 1). This hierarchy-aware approach allows fine-grained teachers to override coarse classifications when confident, while maintaining biological consistency.

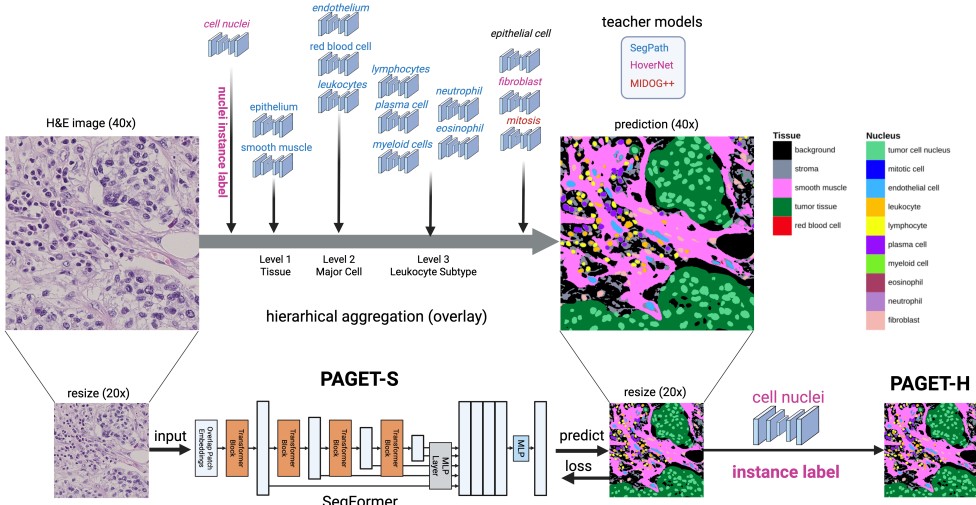

Figure 1: PAGET framework overview.

### 3.4.1. Three-Level Biological Hierarchy

We develop a three-level hierarchical classification scheme that reflects the natural taxonomy of cellular components:

**Level 1 - Tissue Context:** Distinguishes major tissue types (smooth muscle, epithelial tissue), providing essential structural context via SegPath tissue model.

**Level 2 - Major Cell Categories:** Within tissue regions, identifies broad cell types—leukocytes, endothelial cells, red blood cells, and epithelial cells—using SegPath nucleus models.

**Level 3 - Leukocyte Subtypes:** Subdivides leukocytes into lymphocytes, plasma cells, eosinophils, neutrophils and other myeloid cells.

This hierarchy progresses from morphologically stable features to specific phenotypes that require specialized IHC-trained teachers.

### 3.4.2. AGGREGATION ALGORITHM

The aggregation proceeds in five steps. Let $\Omega \subset \mathbb{Z}^2$ denote the image domain and $S(p)$ the final label at pixel $p \in \Omega$.

**Step 1 (Spatial initialization):** HoverNet extracts nucleus instances $\mathcal{N} = \{\mathcal{N}_i \subset \Omega \mid i = 1, \ldots, I\}$, and Otsu thresholding on Gaussian-smoothed images identifies background pixels $\mathcal{B} \subset \Omega$. The remaining pixels form the tissue region $\Omega' = \Omega \setminus (\mathcal{N} \cup \mathcal{B})$.

**Step 2 (Tissue classification):** For each pixel $p \in \Omega'$, we assign a tissue label based on SegPath tissue model logits $\ell_c^{\text{tis}}(p)$ for $c \in \{\text{smooth muscle}, \text{epithelium}\}$:

$$t(p) = \begin{cases} \arg\max_c \ell_c^{\text{tis}}(p), & \max_c \ell_c^{\text{tis}}(p) > 0, \\ \text{stroma}, & \text{otherwise.} \end{cases} \tag{1}$$

**Step 3 (Hierarchical nucleus classification):** Classification proceeds in two stages. First, at the pixel level, teacher predictions are aggregated following the three-level hierarchy (Section 3.4.1): predictions from deeper levels override coarser classifications when teachers at those levels produce positive outputs. For example, a pixel predicted as leukocyte (Level 2) is overridden by lymphocyte (Level 3) if the lymphocyte teacher fires.

Second, at the nucleus level, we aggregate pixel predictions within each nucleus $\mathcal{N}_i$. Let $\text{count}_c(i) = |\{p \in \mathcal{N}_i : s(p) = c\}|$ denote the pixel count for class $c$. We compute the most frequent non-background class:

$$c^*(i) = \arg\max_{c \neq 0} \text{count}_c(i).$$

When $c^*(i) = \text{LEU}$ (leukocyte, Level 2), we further refine to Level 3 subtypes:

$$y(i) = \begin{cases} \arg\max_{s \in \mathcal{S}_{\text{LEU}}} \text{count}_s(i), & \text{if } \exists s \in \mathcal{S}_{\text{LEU}} : \text{count}_s(i) > 0, \\ \text{LEU}, & \text{otherwise,} \end{cases}$$

where $\mathcal{S}_{\text{LEU}} = \{\text{lymphocyte}, \text{plasma cell}, \text{myeloid cell}, \text{eosinophil}, \text{neutrophil}\}$. For all other classes, $y(i) = c^*(i)$.

**Step 4 (Label Completion):** This step completes the label space by addressing cell types that no specialized teacher can directly predict, using two rules: (a) unclassified nuclei within epithelial tissue regions are labeled as epithelial cells, as they predominantly represent epithelial nuclei; (b) nuclei classified only as "connective" by HoverNet are assigned as fibroblasts, since endothelial cells have already been identified by SegPath and fibroblasts constitute the remaining stromal cell population.

**Step 5 (Mitosis integration):** MIDOG++ detections are converted to circular regions of interest (radius 30 pixels), and overlapping nuclei are reclassified as mitotic figures.

This hierarchy-aware aggregation yields the final segmentation $S$ by combining tissue- and nucleus-level decisions in a biologically consistent manner.

### 3.5. Student Model Architecture

We employ SegFormer (Xie et al., 2021) with MiT-B5 encoder (pretrained on ImageNet) for the student model. Input images are processed at 20× magnification (384×384 pixels); many clinical sites operate at 20× due to storage and scanning time constraints, making this resolution practically relevant.

The student directly predicts pixel-wise labels for all 14 classes (13 TME components plus background). We provide two inference variants. **PAGET-S** (Semantic) outputs these pixel-wise predictions directly, optimizing for speed. **PAGET-H** (Panoptic) combines PAGET-S predictions with HoverNet nucleus instance masks, assigning each nucleus the majority class among its constituent pixels. Both variants share the same trained SegFormer weights; PAGET-H adds HoverNet inference time to provide instance-level output.

### 3.6. Training Details

We employ AdamW optimizer with learning rate 6e-5, betas (0.9, 0.999), and weight decay 0.01. The learning rate schedule combines linear warmup from 0 to 1500 iterations followed by polynomial decay from 1500 to 48,000 iterations. Training uses standard CrossEntropyLoss against aggregated teacher labels with batch size 4. Data augmentation includes random resizing (0.85-1.15), cropping (384×384), horizontal/vertical flipping (p=0.5), random blur, gamma adjustment, and photometric distortions. Training was conducted on 8× NVIDIA H100 80GB GPUs.

## 4. Experimental Setup

### 4.1. Datasets

For internal testing, we held out 3,133 images from the training set, covering all 22 cancer types. Here, aggregated teacher predictions serve as ground truth, enabling evaluation of how well the student reproduces teacher supervision.

For external validation, we employed three datasets with human annotations as ground truth. PanopTILs (Liu et al., 2024) provides breast cancer samples with expert annotations. Lizard (Graham et al., 2021) contains colorectal cancer images from four subsets (DigestPath, GlaS, CoNSeP, CRAG); we exclude PanNuke to avoid data leakage, as HoverNet in our teacher ensemble was trained on this dataset. KCCRC is a multi-institutional cohort from Japanese Red Cross Medical Center and Kanagawa Cancer Center, containing colon and gastric samples with pathologist annotations for immune cell subtypes and endothelial cells.

This study was conducted in accordance with the Declaration of Helsinki and approved by the Institutional Review Boards of The University of Tokyo (approval numbers 2381 and 2019158NI), Japanese Red Cross Medical Center (approval number 1414), and Kanagawa Cancer Center (approval number 2020-118).

### 4.2. Baselines

We compare against publicly available representative methods. HD-YOLO (Rong et al., 2023) applies object detection for cell identification (lung and breast variants). Hover-

Net (Graham et al., 2019) provides nuclear instance segmentation and classification (Pan-Nuke and MoNuSAC (Verma et al., 2021) variants). Cerberus (Graham et al., 2023) performs multi-task segmentation; it is excluded from Lizard evaluation due to training set overlap. We also compare against our teacher ensemble to evaluate whether the distilled student can match or exceed teacher performance.

### 4.3. Evaluation Metrics

Due to varying class definitions across datasets and models, we designed hierarchical class mapping in consultation with pathologists (Appendix E). For tissue-level segmentation, we report Dice score. For nucleus-level classification, we report Matthews Correlation Coefficient (MCC) (Chicco et al., 2021) computed per nucleus instance for each class separately. MCC ranges from $-1$ (complete disagreement) to $+1$ (perfect agreement), with 0 indicating random prediction; it provides balanced evaluation for imbalanced classes common in histopathology.

## 5. Results and Discussion

### 5.1. Internal Validation

On internal test data, aggregated teacher predictions serve as ground truth, enabling evaluation of how faithfully the student reproduces teacher supervision. Table 4 (Appendix C) summarizes the results.

Both PAGET-S and PAGET-H achieve high fidelity to teacher labels. For tissue-level segmentation, both variants perform comparably, with IoU scores exceeding 0.70 for stroma and 0.80 for epithelium and smooth muscle. For nucleus-level segmentation, PAGET-H consistently outperforms PAGET-S across all classes except for endothelial cells. For example, epithelial cell nucleus IoU improves from 0.760 to 0.853, and lymphocyte from 0.646 to 0.753. This gain likely stem from majority voting within each nucleus instance, which reduces pixel-level noise in semantic predictions and yields more stable class assignments.

### 5.2. Ablation Study

To validate our hierarchical aggregation design, we compared two strategies using Panop-TILs and KCCRC, which provide ground truth annotations compatible with the $40\times$ resolution at which our IHC-restaining teacher models operate. We evaluated: (1) **flat aggregation**, where the class with maximum logit across all 9 directly-predicted cell types is selected, and (2) **hierarchical aggregation**, using our proposed biological hierarchy, including both partial hierarchies that exclude deeper levels and the full hierarchy. In flat aggregation, all teacher predictions compete directly in a single label space, whereas hierarchical aggregation applies predictions sequentially following biological hierarchy, such that coarse-level decisions constrain downstream subtype classification (Figure 6). The 9 cell types exclude stroma, epithelial cell nuclei, fibroblasts, and mitotic cells, which are assigned through refinement rules rather than direct SegPath prediction (Section 3.4). Both strategies use identical teacher models; only the aggregation method differs.

Table 1 summarizes results. Hierarchical aggregation and its intermediate variants with reduced hierarchy depth consistently outperforms flat aggregation for cell-level classification,

with substantial relative improvements for lymphocytes (+24.5% in PanopTILs, +25.7% in KCCRC) and eosinophils (+81.9% in KCCRC). Performance generally improves as deeper hierarchy levels are incorporated, with gains most evident for cell types at deeper hierarchy levels, where coarse-level context helps disambiguate fine-grained subtypes. Tissue-level segmentation shows comparable performance between strategies, as expected since tissue classification occurs at the first hierarchy level without subsequent refinement.

Table 1: Ablation study: Flat vs hierarchical aggregation with different hierarchy depths (Dice score). Best in **bold**.

| Dataset | Strategy | Epi | Blood | Lym | Pls | Leu |
|---------|----------|-----|-------|-----|-----|-----|
| PanopTILs | Flat | **0.736** | **0.372** | 0.094 | 0.073 | **0.372** |
| | 2 layers (merge level 1 and 2) | **0.736** | **0.372** | 0.117 | **0.118** | 0.369 |
| | Full hierarchy | 0.735 | 0.367 | **0.117** | **0.118** | 0.370 |
| | | **Lym** | **Pls** | **Mye** | **Neu** | **Eos** |
| KCCRC | Flat | 0.358 | 0.017 | 0.223 | 0.205 | 0.149 |
| | 2 layers (merge level 1 and 2) | **0.450** | **0.020** | **0.235** | 0.161 | 0.233 |
| | Full hierarchy | **0.450** | **0.020** | **0.235** | **0.237** | **0.271** |

### 5.3. External Validation

Figure 2 and 7 (Appendix F) shows representative qualitative results on PanopTILs, demonstrating that our proposed variants achieve high-quality results, with PAGET-H providing particularly accurate nuclear boundaries.

Figures 3 and 4 summarize quantitative performance across datasets and cell types. Complete numerical results are provided in Table 6 (Appendix F). Across most cohorts, PAGET-S and PAGET-H consistently outperform both the full teacher pipeline (including HoverNet and refinement rules) and conventional nucleus segmentation models. On KC-CRC, collected from Japanese institutions as was our training data, student and teacher performance are comparable. On datasets from different countries, however, the distilled student frequently exceeds teacher performance, suggesting that the combination of hierarchical aggregation and data augmentation provides effective regularization against distribution shift.

Cell-type-wise comparisons show that PAGET achieves competitive or superior performance across evaluated classes. While baseline models typically support only a subset of cell types, PAGET provides predictions for all 13 TME components from a single model. Notably, higher-level categories such as leukocytes consistently achieve higher accuracy than fine-grained subtypes (Table 6), empirically supporting the assumption underlying our hierarchical aggregation design (Section 3.4).

To evaluate zero-shot generalization capability, we applied PAGET to adenoid cystic carcinoma, a cancer type completely absent from the 22 cancer types in our training dataset (Fig 8) We analyzed one case and had a board-certified pathologist review the segmentation results for obvious errors and missed detections. Notably, tumor cells—which we hypothesized would be the most challenging to generalize given their type-specific morphological characteristics—showed no obvious misclassifications. While some missed detections were

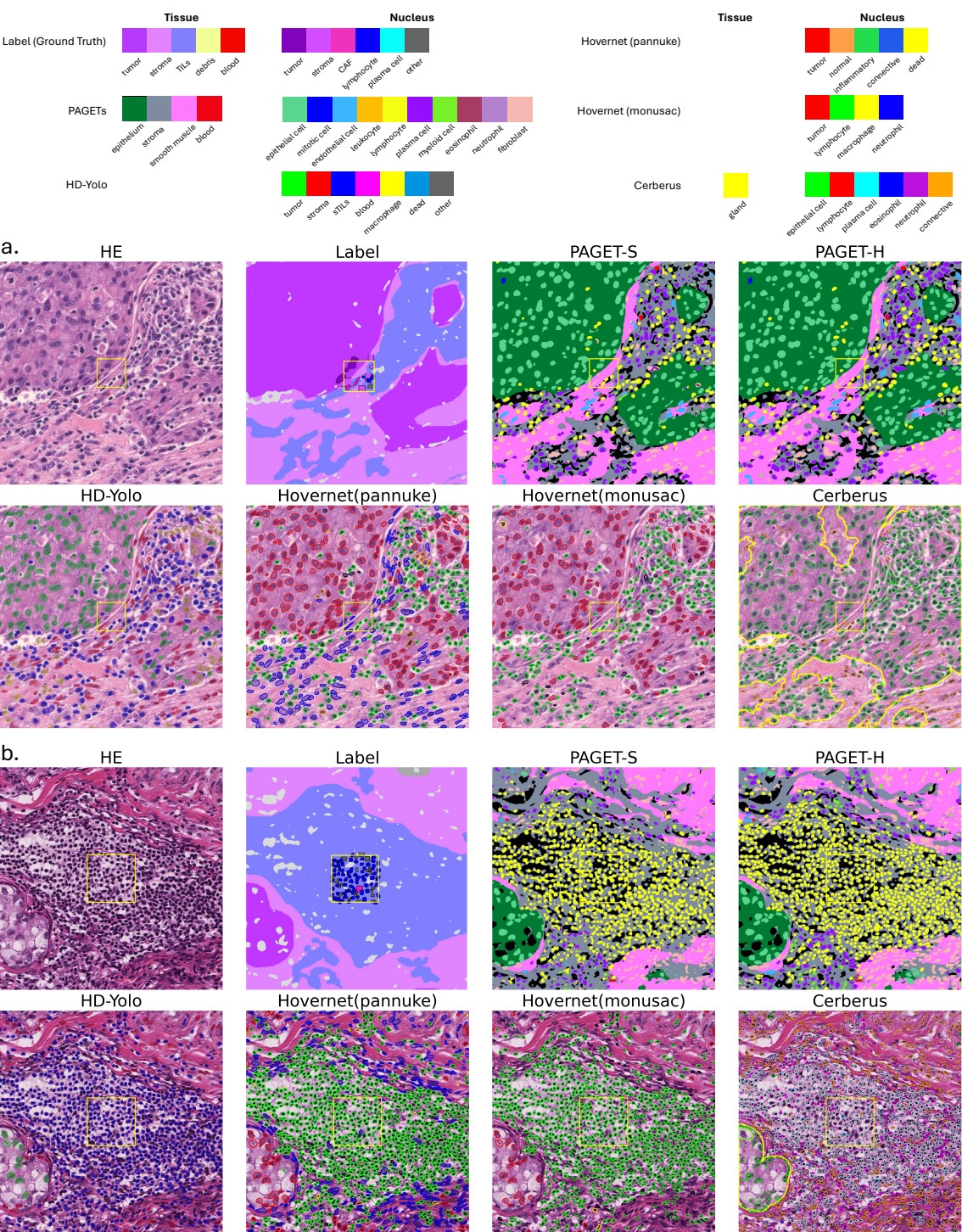

Figure 2: Representative segmentation results on PanopTILs dataset comparing PAGET variants with baseline methods.

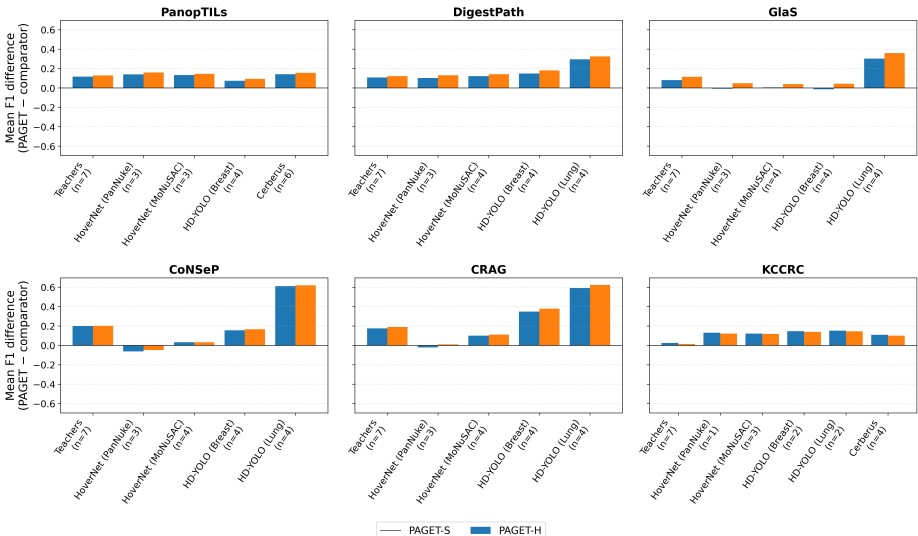

Figure 3: Dataset-wise comparison of PAGET versus baseline models. Bars show mean performance differences (PAGET - baseline), measured by Dice for tissue, MCC for nuclei, computed separately for each dataset. Baselines as indicated on x-axis.

Positive values indicate better performance for PAGET. The number above each bar denotes the number of tissue or nucleus categories available for that comparison.

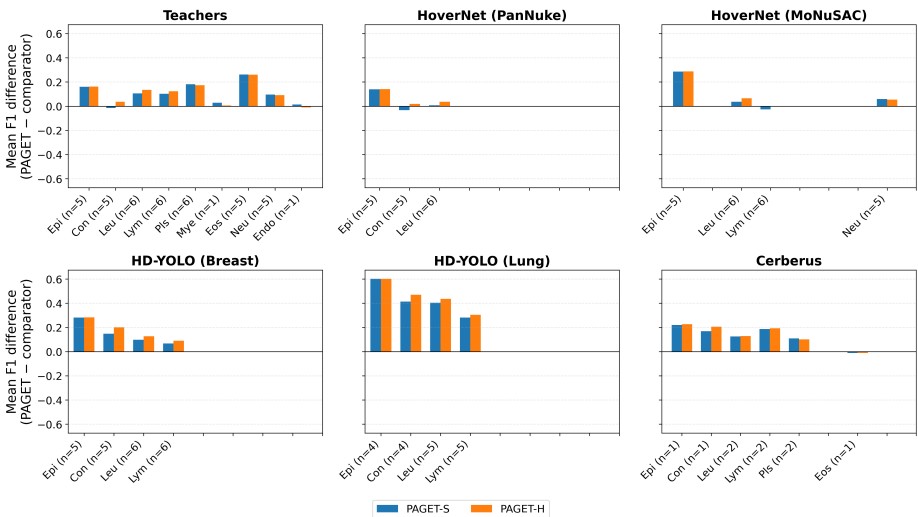

Figure 4: Cell-type-wise comparison of PAGET versus baseline models. Bars show mean performance differences (PAGET - baseline), measured by Dice for tissue, MCC for nuclei, averaged over all datasets in which the corresponding cell type is available. Baselines as indicated on x-axis. Positive values indicate better performance for PAGET. The number above each bar denotes the number of datasets available for that comparison.

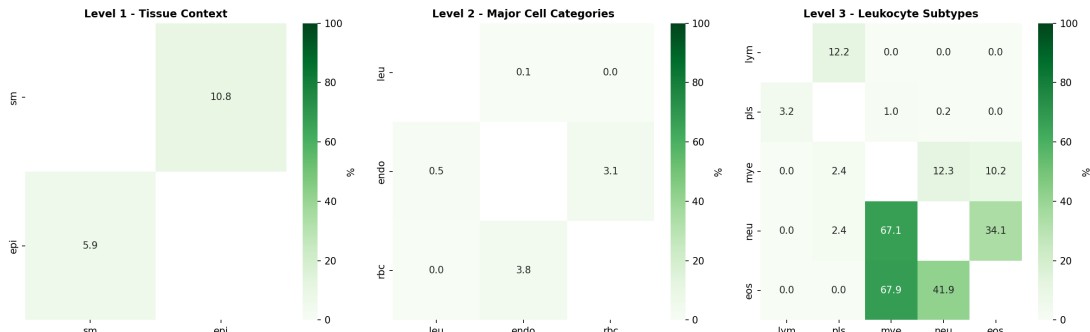

Figure 5: Within-level conflict rates between teacher models. Each cell shows the percentage of high prediction probability pixels (>90%) for the row cell type that also have high probability for the column cell type.

observed, particularly in regions where even the reviewing pathologist found cell type determination ambiguous (marked separately), no cell type showed systematic failure compared to the performance on cancer types included in training. Although this evaluation is limited to a single unseen cancer type and thus definitive conclusions cannot be drawn, these preliminary results suggest that PAGET may have learned generalizable morphological features for cell type recognition that extend beyond the specific cancer types in the training data.

### 5.4. Conflict Cases

To understand failure cases and how hierarchical aggregation handles conflicting predictions, we analyzed pixels where multiple teachers predicted >90% probability simultaneously on the KCCRC dataset (Figure 5). At the tissue level, smooth muscle and epithelium showed moderate overlap (6–11% of high-probability pixels). At the leukocyte subtype level, neutrophil-eosinophil conflicts were frequent (34–42% of high-probability pixels), reflecting the morphological similarity between these granulocytes in H&E. Lymphocyte-plasma cell conflicts were less common (3–12%). When conflicts occur, they typically involve cells that are morphologically ambiguous (Figure 9). Notably, 67–68% of granulocyte high-probability pixels also showed high myeloid probability. Existing models also show limited accuracy for these cell subtypes (Table 5), suggesting that this is an inherently challenging task from H&E images. Despite these conflicts, PAGET achieves competitive or superior performance compared to existing models across evaluated cell types.

### 5.5. Computational Efficiency

PAGET-S processes a 384×384 tile in 4 ms on a single NVIDIA V100 GPU, achieving approximately 207× speedup compared to the teacher ensemble alone and 301× speedup compared to the full teacher pipeline with HoverNet (Table 7, Appendix H). For a typical WSI (100k×100k pixels at 40×), PAGET-S completes processing in approximately 1 minute by directly accessing the 20× layer from the pyramidal image structure, versus over 6 hours

for the full teacher pipeline. PAGET-H, which combines PAGET-S with HoverNet for panoptic segmentation, processes the same WSI in approximately 2 hours, achieving $3.2\times$ speedup.

In terms of accuracy, PAGET-H generally achieves slightly higher performance than PAGET-S, with consistent improvements observed for connective tissue and leukocyte classification (Figure 4). PAGET-S is thus suited for large-scale WSI screening where speed is critical, while PAGET-H is recommended when precise nucleus boundaries are required or when accurate identification of stromal and immune cells is prioritized.

## 6. Conclusion

We presented PAGET, a hierarchical multi-teacher knowledge distillation framework enabling simultaneous segmentation of 13 TME components from H&E slides. Our ablation study validates that aggregating predictions following biological taxonomy improves classification over flat aggregation. The distilled student frequently outperforms the teacher ensemble on external datasets, suggesting effective regularization against distribution shift. PAGET-S and PAGET-H provide a unified solution bridging annotation accuracy and computational efficiency for comprehensive TME characterization.

PAGET relies on the assumption that higher-level tissue and lineage predictions are more reliable than fine-grained subtype predictions. In scenarios where tissue segmentation is unreliable (e.g., under severe domain shift) or when cell subtypes lack clear morphological or immunohistochemical separability in H&E, errors or biases at higher hierarchy levels may propagate to downstream subtype predictions, potentially suppressing correct predictions or amplifying noise introduced during label aggregation.

Beyond the specific teacher models used in this study, the proposed framework is architecture-agnostic and provides a flexible foundation for incorporating future advances, such as foundation-model-based components, as teachers or student backbones (Pachitariu et al., 2025), (Guo et al., 2025), (Hörst et al., 2025)

## Acknowledgments

This work was supported by AMED Practical Research for Innovative Cancer Control grants JP 24ck0106873 and JP 24ck0106904, and JSPS KAKENHI Grant-in-Aid for Scientific Research (B) grant number 21H03836. We thank the pathology teams at collaborating institutions for dataset curation and validation support. We thank Biomy Inc. for developing analysis tools.

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

## Appendix A. Training dataset statistics

Table 2 summarizes the training dataset statistics. Table 3 lists all 22 cancer types with image counts.

Table 2: Comprehensive training dataset statistics

| Nucleus Type | Count | Tissue Type | Pixels |
|---|---|---|---|
| Epithelial cell | 8.4M | Epithelium | 4.0B |
| Fibroblast | 2.9M | Stroma | 2.5B |
| Lymphocyte | 1.2M | Smooth muscle | 2.1B |
| Plasma cell | 357K | Red blood cell | 130M |
| Myeloid cell | 461K | | |
| Eosinophil | 37K | | |
| Neutrophil | 291K | | |
| Endothelial cell | 484K | | |
| Mitotic cell | 2.7K | | |
| **Total nuclei** | **15.4M** | **Total pixels** | **8.7B** |

## Appendix B. Teacher Model Architectural Details

**SegPath Models.** Models for epithelium, smooth muscle, endothelium, red blood cells, and leukocytes follow the architectures and training procedures described in Komura et al. (2023). All models take inputs at $40\times$ magnification.

**Neutrophil Model.** U-Net with EfficientNet-B1 encoder, noisy student pretraining. Input: $492\times492$ at $40\times$. Trained on MPO antibody staining. We utilized the Dice loss function, achieving a validation Dice score of 0.411.

**Eosinophil Model.** DeepLabV3+ with ResNet34 encoder, pretrained on ImageNet. Input: $492\times492$ at $40\times$. Trained on ECP antibody staining. This model also used Dice loss, resulting in a validation Dice score of 0.299.

**MIDOG++.** RetinaNet-based detector trained on the MIDOG++ challenge dataset (Aubreville et al., 2023). Multi-scanner training for robust mitosis detection.

**HoverNet.** Standard architecture from Graham et al. (2019), trained on PanNuke dataset (Gamper et al., 2020). Provides 6-class nucleus classification and instance segmentation.

## Appendix C. Internal Validation

Table 4 summarizes the internal test performance.

## Appendix D. Ablation Study

Figure 6 illustrates the flat and hierarchical aggregation used in the ablation study.

Table 3: Training dataset composition by cancer type

| Cancer Type | Images |
|---|---|
| Endometrial cancer | 3,347 |
| Breast cancer | 3,264 |
| Bladder cancer | 2,884 |
| Urothelial tumor | 2,873 |
| Prostate cancer | 2,790 |
| Kidney tumor | 2,783 |
| Gastric cancer | 2,679 |
| Extrahepatic bile duct cancer | 2,517 |
| Colorectal cancer | 2,290 |
| Triple-negative breast cancer | 2,046 |
| Esophagogastric junction cancer | 2,035 |
| Gastric cancer lymph node metastasis | 1,911 |
| Lung squamous cell carcinoma | 1,852 |
| Benign breast lesion | 1,831 |
| Pancreatic cancer | 1,785 |
| Hypopharyngeal and laryngeal cancer | 1,747 |
| Hepatocellular carcinoma | 1,723 |
| Cervical squamous cell carcinoma | 1,709 |
| Pancreatic neuroendocrine tumor | 1,675 |
| Liver cancer | 1,220 |
| Thymoma | 667 |
| Ovarian mucinous cystic neoplasm | 575 |
| **Total** | **59,443** |

Table 4: Internal test performance (IoU). Best results in **bold**.

| Tissue Class | PAGET-S | PAGET-H | Nucleus Class | PAGET-S | PAGET-H |
|---|---|---|---|---|---|
| Background | 0.847 | **0.848** | Epithelial cell | 0.760 | **0.853** |
| Stroma | 0.709 | **0.715** | Fibroblast | 0.613 | **0.649** |
| Smooth muscle | **0.822** | 0.814 | Mitotic cell | 0.302 | **0.382** |
| Epithelium | 0.772 | **0.809** | Lymphocyte | 0.646 | **0.753** |
| Red blood cell | **0.805** | 0.783 | Plasma cell | 0.556 | **0.612** |
| | | | Myeloid cell | 0.399 | **0.450** |
| | | | Eosinophil | 0.440 | **0.525** |
| | | | Neutrophil | 0.538 | **0.605** |
| | | | Endothelial cell | **0.585** | 0.532 |

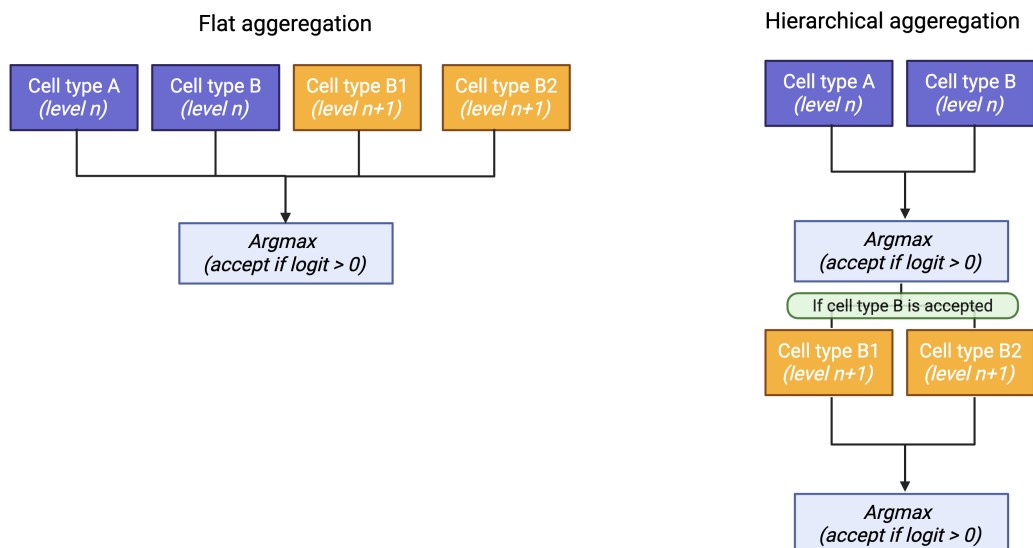

Figure 6: Flat and hierarchical aggregation used in the ablation study.

## Appendix E. Class Mapping for Evaluation

Different datasets and models use varying class definitions. To enable fair comparison, we designed hierarchical class mapping in consultation with pathologists. Table 5 shows the correspondence used for evaluation. For example, PAGET's lymphocyte, plasma cell, myeloid cell, eosinophil, and neutrophil predictions are combined when comparing against ground truth labeled simply as "leukocyte."

## Appendix F. External Validation Results

Table 6 reports comprehensive external validation results across all datasets.

## Appendix G. Conflict cases

Figure 9 shows representative examples of prediction conflicts between teacher models.

## Appendix H. Benchmark Configuration and Processing Time Breakdown

All timing measurements were performed on a single NVIDIA Tesla V100-SXM2-32GB GPU. Each measurement was averaged over 3 runs after 1 warmup iteration using 1,000 patches to minimize model loading overhead. PAGET-S and teacher models were evaluated on equivalent tissue areas ($384 \times 384$ at $20\times$ for PAGET-S, $768 \times 768$ at $40\times$ for teacher models/HoverNet).

Table 7 shows processing time per tile for each pipeline component.

Table 5: Complete class correspondence across evaluation datasets

| Dataset | Eval Class | Class in Dataset | PAGET | HoverNet (PanNuke) | HoverNet (MoNuSAC) | HID-YOLO | Cerberus |
|---|---|---|---|---|---|---|---|
| PanopTILs | Epithelial tissue | cancerous epithelium, normal epithelium, cancer nucleus, normal epithelial nucleus | epi, epi.n | – | – | – | – |
| | Epithelial cell | cancer nucleus, normal epithelial nucleus | epi.n | neopla, no-neo | epi | tumor | epithelial |
| | Connective tissue cell | stromal nucleus, large stromal nucleus | endo, fib | connec | – | stromal | connective tissue cell |
| | Leukocyte | lymphocyte nucleus, plasma cell / large TIL nucleus | lym, pls, mye, eos, neu | inflam | lym, macro, neut | sTILs, macrophage, plasma cell, eosinophil | neutrophil, lymphocyte, plasma cell, eosinophil |
| | Lymphocyte | lymphocyte nucleus | lym | – | lym | sTILs | lymphocyte |
| | Plasma cell | plasma cell / large TIL nucleus | pls | – | – | – | plasma cell |
| Lizard | Epithelial cell | epithelial | epi.n | neopla, no-neo | epi | tumor | – |
| | Connective tissue cell | connective | endo, fib | connec | – | stromal | – |
| | Leukocyte | lymphocyte, plasma, myeloid, eosinophil, neutrophil | lym, mye, eos, neu | inflam | lym, macro, neut | sTILs, macrophage | – |
| | Lymphocyte | lymphocyte | lym | – | lym | sTILs | – |
| | Plasma cell | plasma | pls | – | – | – | – |
| | Eosinophil | eosinophil | eos | – | – | – | – |
| | Neutrophil | neutrophil | neu | – | neutrophil | – | – |
| KCCRC | Endothelial cell | endothelial cell | endo | – | – | – | – |
| | Leukocyte | lymphocyte, plasma cell, myeloid cell, eosinophil, neutrophil | lym, pls, mye, eos, neu | inflam | lym, macro, neut | sTILs, macrophage | neutrophil, lymphocyte, plasma cell, eosinophil |
| | Lymphocyte | lymphocyte | lym | – | lym | sTILs | lymphocyte |
| | Plasma cell | plasma cell | pls | – | – | – | plasma cell |
| | Myeloid cell | myeloid cell, eosinophil, neutrophil | mye | – | – | – | – |
| | Eosinophil | eosinophil | eos | – | – | – | eosinophil |
| | Neutrophil | neutrophil | neu | – | neutrophil | – | neutrophil |
| | Mitotic cell | mitotic cell | mit | – | – | – | – |

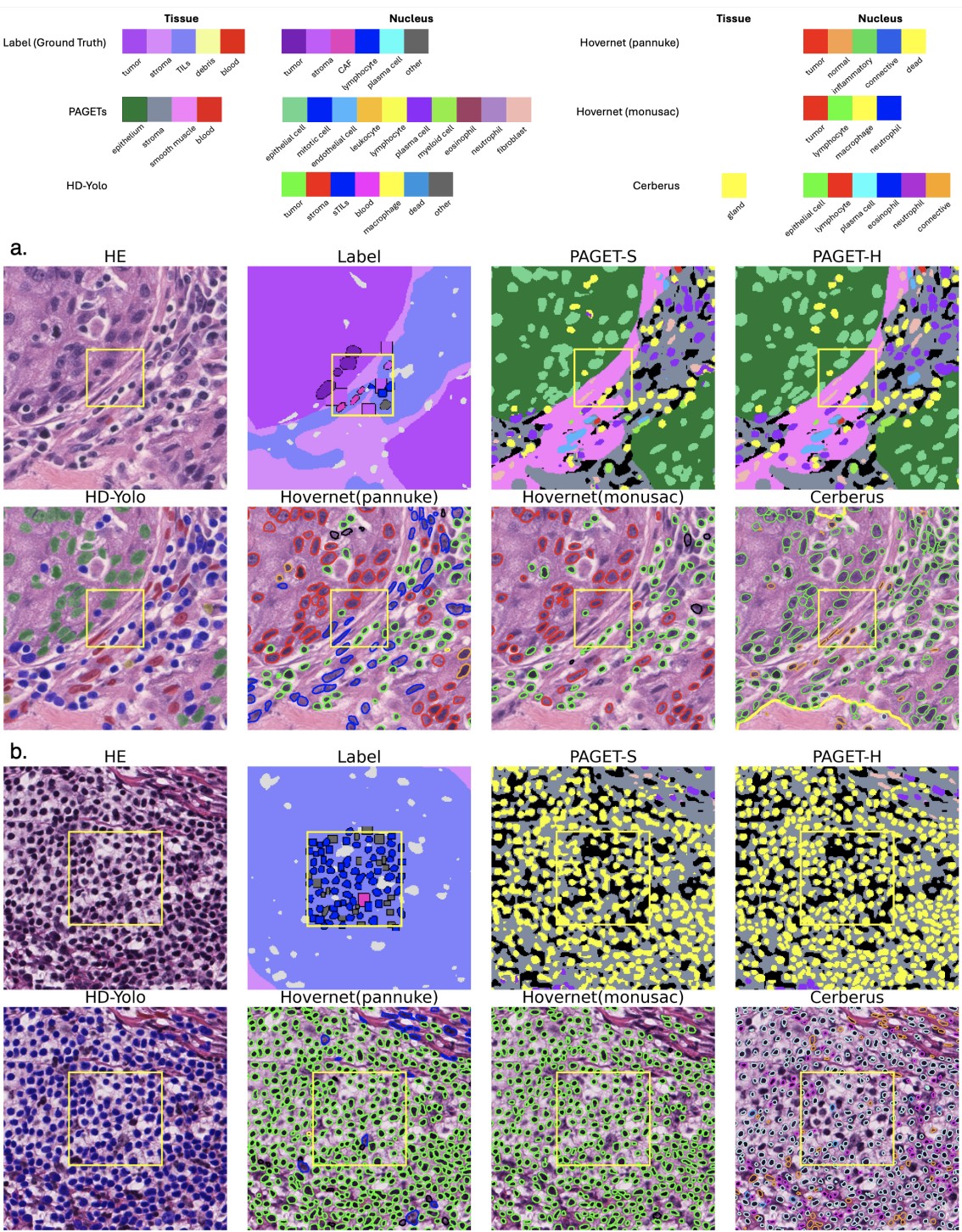

Figure 7: Zoomed-in views of the central regions shown in Figure 2.

Table 6: Comprehensive external validation results across all datasets. Dice scores for tissue-level segmentation, MCC for nucleus-level classification. Best performance in **bold**, second-best underlined. – indicates unsupported class, n/a indicates ground truth data unavailable.

| Dataset | Model | Tissue (Dice) | | | Nucleus (MCC) | | | | | | | | |
|---|---|---|---|---|---|---|---|---|---|---|---|---|---|
| | | Epi | Str | Bld | Epi | Con | Leu | Lym | Pls | Mye | Eos | Neu | Endo |
| PanopTILs | PAGET-S | 0.868 | 0.694 | 0.443 | 0.657 | 0.378 | 0.562 | 0.406 | 0.228 | n/a | 0.242 | 0.123 | n/a |
| | PAGET-H | **0.868** | **0.696** | 0.450 | 0.663 | 0.415 | 0.579 | 0.424 | 0.226 | n/a | 0.255 | 0.099 | n/a |
| | Teacherfull | 0.797 | — | — | 0.435 | 0.295 | 0.345 | 0.209 | 0.176 | n/a | 0 | 0.016 | n/a |
| | HD-YOLO (Breast) | — | — | **0.465** | 0.512 | 0.389 | 0.449 | 0.356 | — | n/a | — | — | n/a |
| | HoverNet (MoNuSAC) | — | — | — | 0.461 | 0.316 | 0.400 | — | — | n/a | — | 0.048 | n/a |
| | HoverNet (PanNuke) | — | — | — | 0.438 | — | 0.424 | 0.368 | — | n/a | — | 0.016 | n/a |
| | HD-YOLO (Lung) | — | — | — | 0.295 | 0.098 | 0.190 | 0.184 | — | n/a | — | — | n/a |
| | Cerberus | 0.800 | — | — | 0.437 | 0.210 | 0.367 | 0.311 | 0.121 | n/a | — | — | n/a |
| DigestPath | PAGET-S | n/a | n/a | n/a | **0.784** | 0.402 | 0.481 | 0.279 | **0.206** | n/a | 0.158 | 0.009 | **0.123** |
| | PAGET-H | n/a | n/a | n/a | 0.777 | **0.423** | 0.550 | 0.320 | 0.192 | n/a | 0.168 | 0.020 | 0.099 |
| | Teacherfull | n/a | n/a | n/a | 0.663 | **0.556** | **0.601** | 0.426 | 0.302 | n/a | 0 | 0 | 0.255 |
| | HoverNet (PanNuke) | n/a | n/a | n/a | 0.539 | 0.376 | 0.480 | 0.234 | 0.187 | n/a | 0 | 0.016 | n/a |
| | HoverNet (MoNuSAC) | n/a | n/a | n/a | 0.476 | 0.402 | 0.482 | 0.272 | 0.033 | n/a | 0.045 | — | n/a |
| | HD-YOLO (Lung) | n/a | n/a | n/a | 0.236 | — | 0.446 | 0.449 | — | n/a | — | 0.048 | n/a |
| | HD-YOLO (Breast) | n/a | n/a | n/a | 0.424 | 0.281 | 0.354 | 0.190 | 0.184 | n/a | — | 0.016 | n/a |
| GlaS | PAGET-S | n/a | n/a | n/a | 0.767 | 0.454 | 0.541 | 0.371 | 0.424 | n/a | 0.470 | 0.378 | n/a |
| | PAGET-H | **0.779** | n/a | n/a | 0.779 | 0.609 | 0.601 | 0.426 | **0.429** | n/a | 0.450 | 0.367 | n/a |
| | Teacherfull | n/a | n/a | n/a | 0.719 | 0.630 | 0.746 | 0.657 | 0.033 | n/a | 0 | 0.249 | n/a |
| | HoverNet (PanNuke) | n/a | n/a | n/a | 0.687 | 0.498 | 0.407 | 0.234 | 0.000 | n/a | 0.000 | — | n/a |
| | HoverNet (MoNuSAC) | n/a | n/a | n/a | 0.525 | 0.505 | 0.597 | 0.513 | — | n/a | — | 0.084 | n/a |
| | HD-YOLO (Lung) | n/a | n/a | n/a | 0.372 | 0.143 | 0.208 | 0.204 | — | n/a | — | 0.045 | n/a |
| | HD-YOLO (Breast) | n/a | n/a | n/a | 0.638 | 0.409 | 0.613 | 0.526 | — | n/a | — | — | n/a |
| CoNSeP | PAGET-S | n/a | n/a | n/a | 0.894 | 0.611 | 0.712 | 0.518 | 0.424 | n/a | 0.470 | 0.374 | n/a |
| | PAGET-H | n/a | n/a | n/a | **0.904** | **0.695** | 0.737 | 0.546 | **0.429** | n/a | 0.450 | 0.367 | n/a |
| | Teacherfull | n/a | n/a | n/a | 0.710 | 0.688 | 0.563 | 0.335 | 0.108 | n/a | 0.000 | 0.249 | n/a |
| | HoverNet (PanNuke) | n/a | n/a | n/a | 0.860 | **0.736** | **0.831** | — | 0.000 | n/a | 0.000 | — | n/a |
| | HoverNet (MoNuSAC) | n/a | n/a | n/a | 0.714 | — | 0.771 | **0.674** | — | n/a | — | **0.386** | n/a |
| | HD-YOLO (Lung) | n/a | n/a | n/a | 0.129 | 0.157 | 0.081 | 0.094 | — | n/a | — | — | n/a |
| | HD-YOLO (Breast) | n/a | n/a | n/a | 0.619 | 0.382 | 0.688 | 0.589 | — | n/a | — | — | n/a |
| CRAG | PAGET-S | n/a | n/a | n/a | **0.877** | 0.611 | 0.507 | 0.539 | **0.498** | **0.343** | 0.331 | 0.343 | n/a |
| | PAGET-H | n/a | n/a | n/a | 0.864 | **0.695** | 0.497 | 0.533 | 0.484 | 0.320 | 0.331 | 0.340 | n/a |
| | Teacherfull | n/a | n/a | n/a | 0.772 | 0.688 | 0.487 | — | 0.477 | 0.314 | 0.267 | 0.150 | n/a |
| | HoverNet (PanNuke) | n/a | n/a | n/a | 0.794 | 0.663 | 0.377 | — | 0.108 | — | 0 | 0.000 | n/a |
| | HoverNet (MoNuSAC) | n/a | n/a | n/a | 0.637 | — | **0.808** | **0.619** | — | — | — | 0.065 | n/a |
| | HD-YOLO (Lung) | n/a | n/a | n/a | 0.121 | 0.025 | 0.095 | 0.115 | — | — | — | 0.150 | n/a |
| | HD-YOLO (Breast) | n/a | n/a | n/a | 0.375 | 0.257 | 0.383 | 0.315 | — | — | — | — | n/a |
| KCCRC | PAGET-S | n/a | n/a | n/a | n/a | n/a | **0.507** | 0.539 | **0.498** | **0.343** | 0.331 | 0.385 | 0.251 |
| | PAGET-H | n/a | n/a | n/a | n/a | n/a | 0.497 | 0.533 | 0.484 | 0.320 | 0.331 | **0.388** | 0.226 |
| | Teacherfull | n/a | n/a | n/a | n/a | n/a | 0.487 | **0.600** | 0.477 | 0.314 | 0.267 | 0.296 | 0.237 |
| | HoverNet (PanNuke) | n/a | n/a | n/a | n/a | n/a | 0.377 | 0.307 | — | — | 0 | 0.357 | — |
| | HoverNet (MoNuSAC) | n/a | n/a | n/a | n/a | n/a | 0.404 | 0.307 | — | — | — | — | — |
| | HD-YOLO (Lung) | n/a | n/a | n/a | n/a | n/a | 0.378 | 0.364 | — | — | — | — | — |
| | HD-YOLO (Breast) | n/a | n/a | n/a | n/a | n/a | 0.463 | 0.291 | — | — | — | — | — |
| | Cerberus | n/a | n/a | n/a | n/a | n/a | 0.453 | 0.261 | 0.387 | — | **0.343** | — | — |

*Abbreviations:* Epi=Epithelium/Epithelial, Str=Stroma, Bld=Blood, Con=Connective tissue, Leu=Leukocyte, Lym=Lymphocyte, Pls=Plasma cell, Mye=Myeloid cell, Eos=Eosinophil, Neu=Neutrophil, Endo=Endothelial cell, Fib=Fibroblast, Mit=Mitotic cell

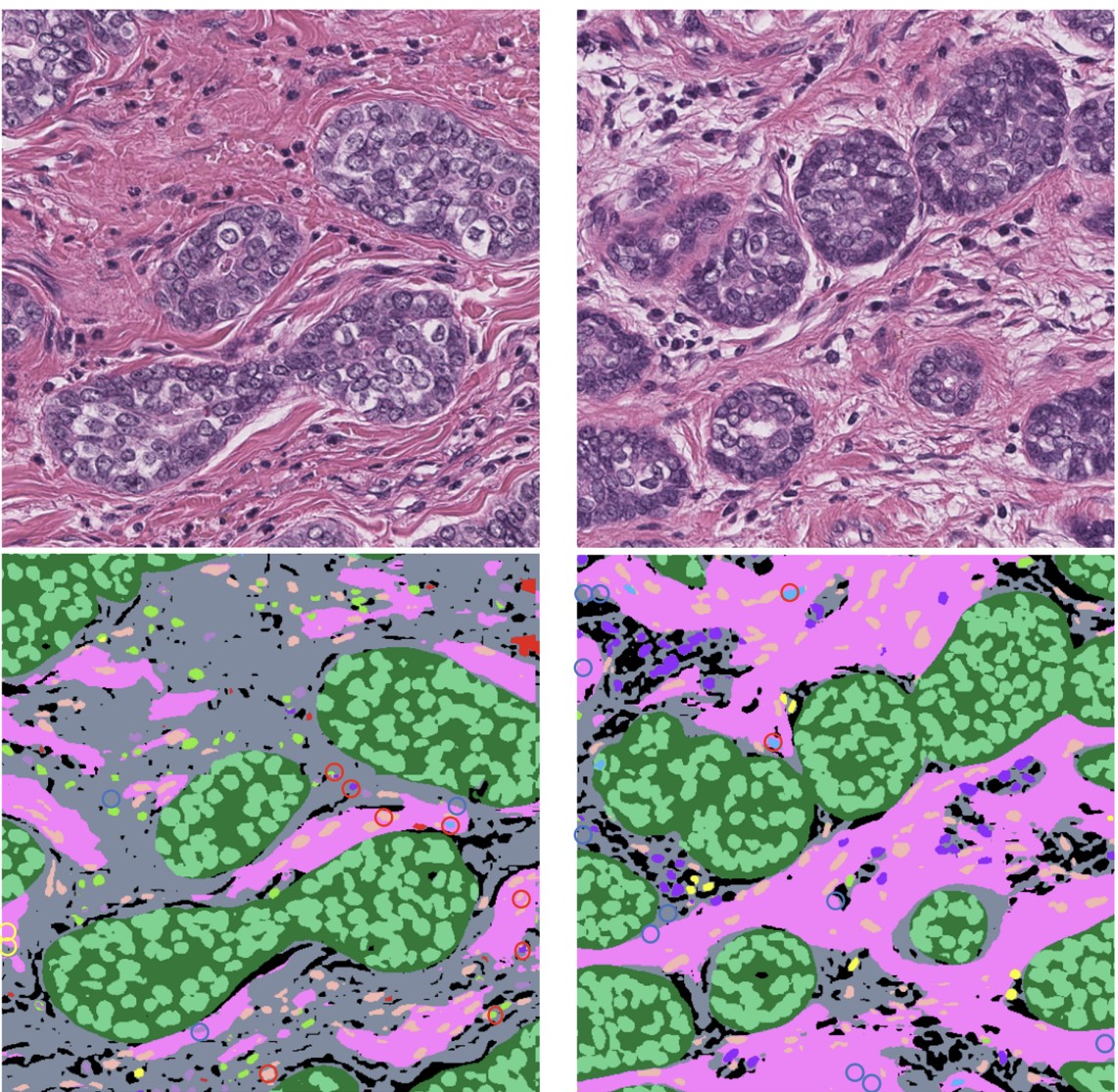

Figure 8: Zero-shot generalization to an unseen cancer type. Representative regions from an adenoid cystic carcinoma case, a cancer type not included in the training data, showing H&E-stained images (top) and corresponding PAGET-S segmentation results (bottom). A board-certified pathologist reviewed the segmentation outputs and annotated errors: red circles indicate obvious misclassifications, blue circles indicate obvious missed detections, and yellow circles indicate missed cells where the cell type was ambiguous even upon expert review.

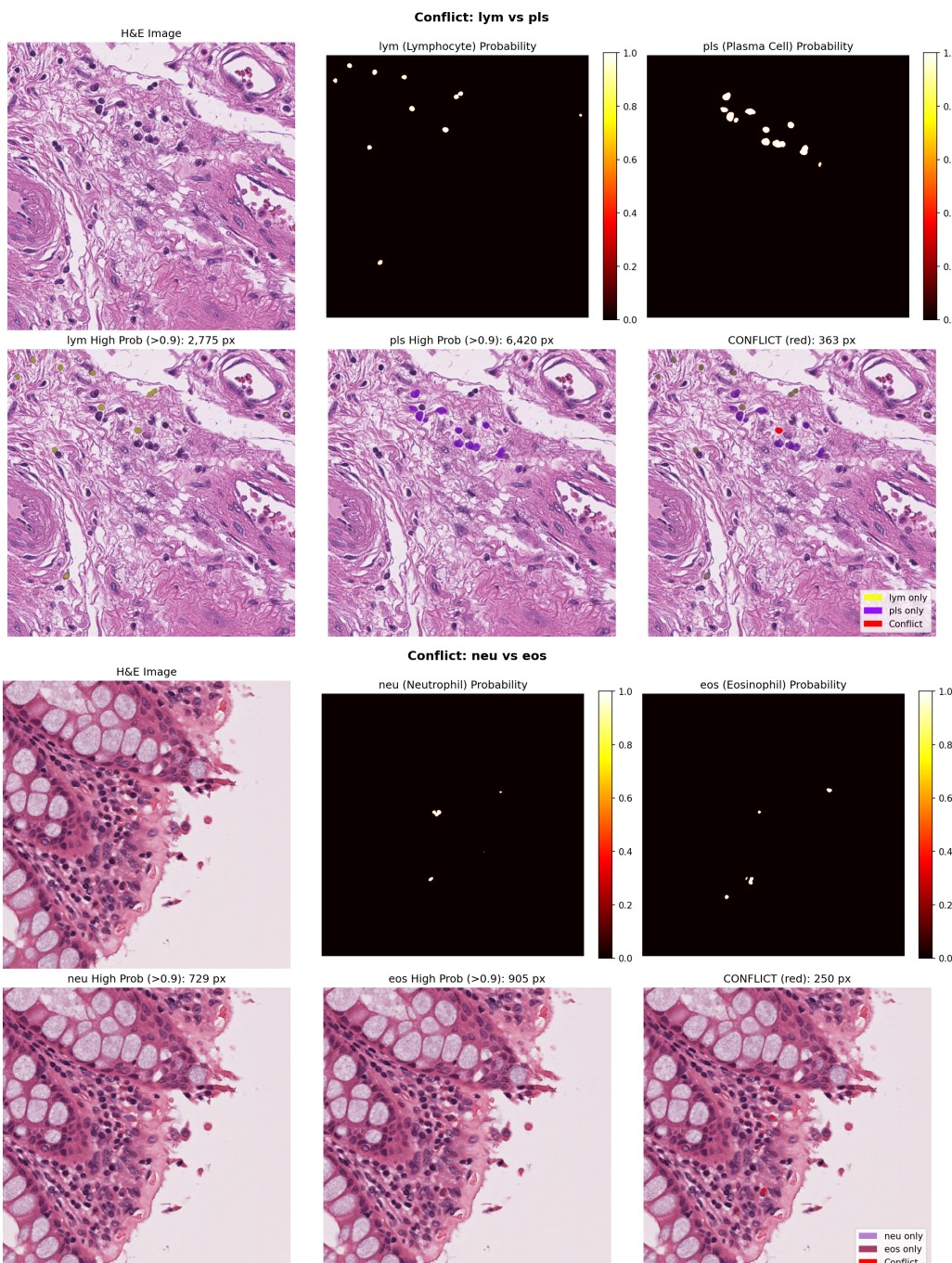

Figure 9: Representative examples of prediction conflicts between teachers. Top: lymphocyte-plasma cell conflicts. Bottom: neutrophil-eosinophil conflicts. Each panel shows the H&E image, probability maps for each class, pixels with over 90% prediction probability for each class, and conflict regions where both classes exceed 90% probability (red).

Table 7: Processing time per tile (ms) on Tesla V100-SXM2-32GB.

| Component | Input Size | Time (ms) | Speedup |
|---|---|---|---|
| *Individual components* | | | |
| PAGET-S (SegFormer) | 384×384 @20× | 4.3 ± 0.1 | – |
| HoverNet | 768×768 @40× | 403 ± 1 | – |
| Teacher | 768×768 @40× | 890 ± 23 | – |
| *Combined pipelines* | | | |
| **PAGET-S** (semantic only) | – | **4.3** | 301× vs Teacher$_{full}$ |
| **PAGET-H** (PAGET-S + HoverNet) | – | **407** | 3.2× vs Teacher$_{full}$ |
| Teacher | – | 890 | – |
| Teacher$_{full}$ (Teacher + HoverNet) | – | 1,293 | (reference) |

