# OpenReview forum: "PAGET: Hierarchical Multi-Teacher Knowledge Distillation for Comprehensive Tumor Microenvironment Segmentation"
_MIDL.io/2026/Conference — MIDL 2026 Poster_

### Official Review · Reviewer_Ur6F · 2026-01-04

**Confidence:** 4
**Preliminary Rating:** 5
**Final Rating:** 5

**Summary:**

The authors propose hierarchical multi-teacher knowledge distillation framework designed to segment different TME components from H&E-stained histopathology images. To overcome the limitations of manual annotation and single-teacher models, the method aggregates predictions from specialized teachers that is trained on IHC restaining data, using a biological taxonomy that respects cell lineages. The framework includes semantic (PAGET-S) and panoptic (PAGET-H) variants. Experiments on three external datasets demonstrate that the student model consistently outperforms the teacher ensemble, providing a unified and efficient solution for detailed TME analysis.

**Strengths:**

The hierarchical aggregation strategy is a significant contribution. Unlike standard flat ensembles, this approach aggregates teacher predictions following biological taxonomy which is physically motivated and innovative for this domain.
The paper addresses a major gap in the field by enabling the simultaneous segmentation of 13 distinct TME classes, significantly expanding upon previous methods that typically handle only 3-6 classes.
The authors utilize a massive distillation dataset of about 59k images and validate on three diverse external datasets, demonstrating strong generalization capabilities.
By distilling multiple computationally heavy IHC-trained teachers into a single efficient student model, the work makes large-scale, detailed TME analysis feasible.

**Weaknesses:**

While the student outperforms the teachers, the upper bound of performance is inherently influenced by the quality of the pseudo-labels generated by the teacher ensemble. If the IHC-trained teachers have systematic biases, these could be propagated.
The training pipeline involves managing multiple specialized teachers. While the inference model is efficient, the training complexity is high, which might hinder reproducibility for groups without access to similar teacher resources.
The paper compares the student primarily against the teacher ensemble. Broader comparisons with other recent state-of-the-art semi-supervised or weakly-supervised segmentation methods on H&E data would strengthen the positioning of the method.

**Detailed Comments:**

Please ensure the code release includes the pre-trained teacher weights or the exact specifications to reproduce the pseudo-label generation step.
The distinction between PAGET-S and PAGET-H is mentioned, but a clearer breakdown of the performance trade-offs (speed vs accuracy) between these two variants in the main results section would be beneficial.

**Justification Of Final Rating:**

I have carefully read the authors' rebuttal and official comments, and I find that they have satisfactorily addressed the questions and concerns raised in my initial review. Given the novel hierarchical methodology, the extensive validation across diverse datasets, and the thorough revisions made during the rebuttal phase, I maintain my rating.

**Justification Of The Preliminary Rating:**

This paper presents a high-quality solution to a significant problem in computational pathology that is detailed segmentation of the TME. The method is methodologically sound (hierarchical distillation), technically impressive (integration of multiple teachers), and nicely validated. The shift from flat classification to taxonomy-aware aggregation is a valuable insight that justifies a strong acceptance.

**Questions To Address In The Rebuttal:**

Can you provide more insight into the failure cases? Are there specific rare cell types among the 13 classes where the distillation process struggles?
How does the hierarchical aggregation handle conflicting strong predictions from different teachers that violate the biological taxonomy?

---

> ### Author Response · Authors · 2026-01-25
>
> We thank the reviewer for the thoughtful questions. Our revisions are highlighted in cyan in the revised manuscript.
>
> **Q1. Please ensure the code release includes the pre-trained teacher weights or the exact specifications to reproduce the pseudo-label generation step.**
>
> A1. Thank you for this comment. We confirm that the code for the pseudo-label generation step is publicly available.
>
> **Q2. The distinction between PAGET-S and PAGET-H is mentioned, but a clearer breakdown of the performance trade-offs (speed vs accuracy) between these two variants in the main results section would be beneficial.**
>
> A2. Thank you for this suggestion. We have revised the Computational Efficiency section to explicitly compare PAGET-S and PAGET-H in terms of both speed and accuracy (page 12). PAGET-H generally achieves slightly higher accuracy than PAGET-S, with consistent improvements for connective tissue and leukocyte classification (Figure 4). We now clarify that PAGET-S is suited for large-scale WSI screening where speed is critical, while PAGET-H is recommended when precise nucleus boundaries (described in page 8) are required or when accurate identification of stromal and immune cells is prioritized.
>
> **Q3. Can you provide more insight into the failure cases? Are there specific rare cell types among the 13 classes where the distillation process struggles? How does the hierarchical aggregation handle conflicting strong predictions from different teachers that violate the biological taxonomy?**
>
> A3. Thank you for this insightful question. To address this, we conducted a quantitative analysis of teacher model conflicts on the KCCRC dataset, defining conflicts as pixels where multiple teachers predicted >90% probability simultaneously (Figure 5).
> At the tissue level, smooth muscle and epithelium showed moderate overlap (6–11% of high-probability pixels). At the leukocyte subtype level, neutrophil-eosinophil conflicts were frequent (34–42% of high-probability pixels), reflecting the morphological similarity between these granulocytes in H&E. Lymphocyte-plasma cell conflicts were less common (3–12%). When conflicts occur, they typically involve cells that are morphologically ambiguous (Figure 9). Notably, 67–68% of granulocyte high-probability pixels also showed high myeloid probability.
> Existing models also show limited accuracy for these cell subtypes (Figure 4), suggesting that this is an inherently challenging task from H&E images. Despite these conflicts, PAGET achieves competitive or superior performance compared to existing models across evaluated cell types. We have added this analysis to the revised manuscript (page 12).

---

### Official Review · Reviewer_qBgq · 2026-01-09

**Confidence:** 5
**Preliminary Rating:** 4
**Final Rating:** 5

**Summary:**

This paper proposed PAGET, a hierachical multi-teacher models knowledge distillation framework for comprehensive tumor microenvironment (TME) segmentation from H&E-stained histopathology images. The key innovations lie in utilizing multiple teachers (trained on immunohistochemical restaining data),  and combine with the biological relation (a three-level hierarchical ensemble method) to generate training labels, which will be used for distillating knowledge to the student variants (PAGET-S and PAGET-H). The idea is sound and related to the clinical practice.

**Strengths:**

1. The hierarchical aggregation strategy that respects biological taxonomy is interesting and a reasonable in medical domain as a step to enhance the standard ensemble methods for more robust labels.

2. The training datasets and validations are comprehensive, covering the large-scale publically available annotated datasets.

3. Application to 20x magnification for popular usage.

**Weaknesses:**

1. Limited theoretical analysis. Specially for Section 3.4. (Teacher Prediction Aggregation). It is unclear why and when this approach should outperform alternatives. When this aggregation scheme tend to fail?

2. Limited Ablation study. Some factors are interesting to look at: (1) the impact of each the impact of each hierarchy level independently. (2) the contribution of different teacher models to final performance.

3. Explain or using a figure to depict the hierarchical vs. flat aggregation.

4. The evaliation figures are confusing to readers. For example, Figure 3, 4. what the y axis represent should be explained. mean - diff

**Detailed Comments:**

The idea of this work is sound and the proposed model or datasets are useful for the field. The manuscript could be enhance from better figures, methodologies description and discussion.

**Justification Of Final Rating:**

The authors rebuttal were detailed and the paper quality fullfilled the conference. It answers my asked questions, so I changed the final rating to 5 for this submission. I believed this work could be useful to the community.

**Justification Of The Preliminary Rating:**

This paper is innovative. The provided multi-teacher aggregation framework is useful in medical and clinical domain. It can also be translated to other field that requires large-scale annotated labels.

**Questions To Address In The Rebuttal:**

1. For Section 3.4. (Teacher Prediction Aggregation). It is unclear why and when this approach should outperform alternatives. When this aggregation scheme tend to fail?

2. Limited Ablation study. Some factors are interesting to look at for the ablation study: (1) the impact of each the impact of each hierarchy level independently. (2) the contribution of different teacher models to final performance.

3. Explain and use a figure to depict the hierarchical vs. flat aggregation for better interpretation of the method.

4. The evaliation figures are confusing to readers. For example, Figure 3, 4. what the y axis represent should be explained. mean - diff

5. Figure 2 is hard for visualization (e.g., the yellow box is hard to see in PAGET-H and PAGET-S visualization)

6. Following Figure 2, It seems that the proposed methods provide more detailed predictions (e.g., include the tissue-level outcomes), which could be a benefit of distillating knowledge from multiple teachers. But how about the failure cases? What could lead to bad behaviors because the aggregated labels could  inroduce more bias/noise.

7. Generalization to unseen cancer types: The training data includes 22 cancer types, It would be better to provide zero-shot evaluation on completely unseen cancer types to demonstrate generalization capability?

8. Suggested include into discussion. Can this innovative framework include some foundation models for better performance. Some example of applications are like below.

https://doi.org/10.1101/2025.04.28.651001
https://doi.org/10.1038/s43856-025-01205-x
https://doi.org/10.48550/arXiv.2501.05269

---

> ### Author Response · Authors · 2026-01-25
>
> We thank the reviewer for the detailed feedback. Our revisions are highlighted in magenta in the revised manuscript.
> Thank you.
>
> **Q1. For Section 3.4. (Teacher Prediction Aggregation). It is unclear why and when this approach should outperform alternatives. When this aggregation scheme tend to fail?**
>
> A1. Thank you for raising this important point. To address this, we have clarified the key assumption and scope of applicability of our aggregation strategy in Section 3.4 (page 4) of the revised manuscript. Specifically, we now explicitly state that our approach relies on the assumption that higher-level biological categories are more robust to classify from H&E images than fine-grained subtypes. The hierarchical aggregation exploits this asymmetry by allowing coarse predictions to constrain the decision space before applying more uncertain subtype-level teachers, which explains why the method can outperform flat aggregation when teachers operate at different semantic levels. This assumption is empirically supported by our external validation results (Table 6), where higher-level categories such as leukocytes consistently achieve higher accuracy than fine-grained subtypes. We have added this observation to the External Validation section (page 8) to provide empirical justification for the hierarchical design. We also clarify that this advantage is not universal: the approach may fail when higher-level predictions themselves are unreliable, or when subtypes do not exhibit clear morphological or immunohistochemical separation. These failure cases are now explicitly discussed as limitations in the Conclusion (page 12).
>
> **Q2.Limited Ablation study. Some factors are interesting to look at for the ablation study: (1) the impact of each the impact of each hierarchy level independently. (2) the contribution of different teacher models to final performance.**
>
> A2. (1) Impact of each hierarchy level.
> We agree that this is an important ablation. We therefore conducted an additional analysis in which we selectively removed individual hierarchy levels from the aggregation while keeping the teacher models fixed. Concretely, we evaluated partial hierarchies that exclude deeper levels and compared them against the full hierarchy(page 8).
> The results show that using the full hierarchy yields the best overall performance, whereas removing an intermediate level leads to degraded accuracy. These findings indicate that each hierarchy level contributes complementarily, and that coarse-level context alone is insufficient to recover fine-grained cell identity. We included these results and their interpretation in the revised manuscript.
>
> (2) Contribution of different teacher models.
>  We agree that this is an interesting direction. However, the feasibility of such ablation depends on the teacher's role: some teachers (e.g., HoverNet for instance segmentation, SegPath tissue model for spatial context) are integral to the aggregation pipeline and cannot be simply removed, while removing specialized teachers (e.g., MPO model) would eliminate the corresponding class labels entirely. It would also be valuable to investigate how final performance changes when substituting teachers with alternatives of varying accuracy, particularly at higher hierarchy levels where errors may propagate downstream. Designing fair evaluation protocols for these analyses was beyond the scope of the current rebuttal period, but we recognize this as a valuable direction for future investigation.
>
> **Q3.Explain and use a figure to depict the hierarchical vs. flat aggregation for better interpretation of the method.**
>
> A3. Thank you for the suggestion. We have addressed this point by adding a new schematic figure that explicitly contrasts hierarchical aggregation with flat aggregation, along with an explanatory description, to improve the interpretability of the method in the revised manuscript (page 7 and Figure 6 in page 17).
>
> **Q4.The evaliation figures are confusing to readers. For example, Figure 3, 4. what the y axis represent should be explained. mean - diff**
>
> A4. Thank you for pointing this out. We have revised the captions and axis labels of Figures 3 and 4 to clearly define the y-axis as the mean performance difference (PAGET − baseline), specify the metrics used (Dice for tissue, MCC for nuclei), unify the terminology by consistently using "baseline," and indicate that baselines correspond to models on the x-axis. (page 10)
>
> **Q5.Figure 2 is hard for visualization (e.g., the yellow box is hard to see in PAGET-H and PAGET-S visualization)**
>
> A5.Thank you for pointing this out. To improve visualization, we have added magnified views of the yellow boxed regions as Figure 7 (page 20).

---

> ### Author Response · Authors · 2026-01-25
>
> **Q6.Following Figure 2, It seems that the proposed methods provide more detailed predictions (e.g., include the tissue-level outcomes), which could be a benefit of distillating knowledge from multiple teachers. But how about the failure cases? What could lead to bad behaviors because the aggregated labels could inroduce more bias/noise.**
>
> A6. Thank you for this question. We agree that aggregating predictions from multiple teachers can in principle introduce bias or noise if individual teachers have limited accuracy. However, our teacher ensemble is largely based on IHC-trained models. As noted in the Introduction and demonstrated in our previous work (Komura et al., Patterns 4(2), 2023), morphology-based annotations by pathologists can be inaccurate for cells with atypical morphology, whereas IHC-based annotation captures cells based on molecular markers regardless of morphological appearance, resulting in less biased supervision. Furthermore, our external validation results show that PAGET consistently outperforms existing models across multiple datasets and cell types, providing empirical support that the aggregated labels are of sufficient quality to train a well-generalizing student.
>
> **Q7. Generalization to unseen cancer types: The training data includes 22 cancer types, It would be better to provide zero-shot evaluation on completely unseen cancer types to demonstrate generalization capability?**
>
> A7.Thank you for this suggestion. Our training data already covers a broad range of epithelial malignancies (22 cancer types), which makes it challenging to identify completely unseen epithelial tumor types for systematic evaluation. Nevertheless, we examined adenoid cystic carcinoma, a cancer type completely absent from our training data, as a qualitative zero-shot case study.
> A board-certified pathologist reviewed the predictions. Notably, tumor cells—whose morphology varies most between cancer types—showed no obvious misclassifications. While some missed detections were observed in regions where even the reviewing pathologist found cell type determination ambiguous, no cell type showed systematic failure. Although this evaluation is limited to a single unseen cancer type, these preliminary results suggest that PAGET may have learned generalizable morphological features beyond the specific cancer types in the training data. We have included this analysis in the revised manuscript (page 8, Figure 8).
>
> **Q8.Suggested include into discussion. Can this innovative framework include some foundation models for better performance. Some example of applications are like below.**
>
> A8.Thank you for this suggestion. We agree that the proposed framework is architecture-agnostic and can naturally incorporate foundation models. Accordingly, we added a brief discussion of this potential extension as a future direction in the Conclusion and incorporated citations to the foundation-model-based studies suggested by the reviewer (page 12).

---

### Official Review · Reviewer_naJD · 2026-01-10

**Confidence:** 4
**Preliminary Rating:** 4

**Summary:**

The paper introduces PAGET, a multi-teacher knowledge distillation (KD) framework designed for comprehensive 13-class tumor microenvironment (TME) segmentation from H&E-stained histopathology images. The primary technical contribution is a taxonomy-aware aggregation strategy that organizes specialized teacher outputs into a three-level biological hierarchy.

**Strengths:**

1. The method addresses a major bottleneck in digital pathology: the computational impossibility of running dozens of specialized IHC-trained models on WSIs for comprehensive TME analysis
2. Unlike standard multi-teacher KD that assumes a shared label space, the proposed hierarchical aggregation reflects biological realit

**Weaknesses:**

The student model relies on a standard SegFormer (MiT-B5) encoder. While the distillation framework is novel, the underlying neural architecture contributes little to the state-of-the-art in model design

**Detailed Comments:**

1. The performance gain in PAGET-H over PAGET-S in nucleus-level IoU is a strong result, justifying the extra computation for panoptic segmentation.
2. The speedup of 301x for PAGET-S on WSIs (1 minute vs 6 hours) is highly impressive and makes the tool viable for clinical workflows.
3. The choice of Matthews Correlation Coefficient (MCC) for evaluation is appropriate given the inherent class imbalance in TME component

**Justification Of The Preliminary Rating:**

The paper provides a significant and technically sound contribution to computational pathology by solving the trade-off between segmentation complexity and inference speed. The use of a biological hierarchy for teacher aggregation is a clever and well-justified departure from standard KD techniques. While the architectural novelty is modest, the empirical validation on a large, multi-organ scale and the demonstration of superior generalization on external cohorts make this a valuable paper for the community. The 301x speedup is a compelling "real-world" result.

**Questions To Address In The Rebuttal:**

How do the "Refinement Rules" in Step 4 affect generalization? If these rules were removed, would the student model still outperform the teacher ensemble on external datasets?

---

> ### Author Response · Authors · 2026-01-25
>
> We thank the reviewer for the helpful comment. Our revisions are highlighted in red in the revised manuscript.
>
> **Q1. How do the "Refinement Rules" in Step 4 affect generalization? If these rules were removed, would the student model still outperform the teacher ensemble on external datasets?**
>
> A1. Thank you for the question, and we apologize for the confusion.
> The rules in Step 4 are not performance heuristics but are required to define supervision for cell types that cannot be directly predicted by any teacher model, specifically epithelial cell nuclei and fibroblasts. Without this step, these classes would remain unlabeled in the aggregated pseudo-labels.
> Therefore, removing Step 4 would change the label space itself, making a direct comparison infeasible: a student trained without this step would not be able to recognize epithelial or fibroblast nuclei.
> To clarify this point, we have renamed “Refinement” rules to “Label Completion” rules in the revised manuscript (page 5) and explicitly state their role in completing the label space rather than improving performance.

---

### Author Rebuttal · Authors · 2026-01-25

**Rebuttal:**

We thank all reviewers for their careful reading of our manuscript and their constructive suggestions. We have revised the manuscript accordingly and provide point-by-point responses below. Revisions are color-coded as follows: red (Reviewer naJD), magenta (Reviewer qBgq), and cyan (Reviewer Ur6F).

We are attaching the revised manuscript as the supporting material here.

**Supporting Material:**

/attachment/47cc17577f9d9ce999acb0f4627ff2a9c5deb8d1.pdf

---

### Comment · Area_Chair_cFoE · 2026-01-30
**Reviewers, please take a moment to review the authors' rebuttals and revise your ratings**

Update your final rating by clicking “Edit” → “Official Review” and providing the Final Rating by February 1st 2026 (23:59 AoE).

---

### Meta-Review · Area_Chair_cFoE · 2026-02-09

**Recommendation:** Accept (Oral)
**Confidence:** 5

**Metareview:**

This manuscript presents a multi-teacher knowledge distillation-based approach, hierarchically aggregated using biological taxonomy, to train a single model that simultaneously segments 13 tumor microenvironment components. Reviewers found the strengths of this work to be an innovative hierarchical aggregation strategy, the large reported speedup, and the evaluation on a large, diverse set of images spanning multiple cancer types. The authors were responsive to the reviewers' critiques. A remaining weakness is the comparison with the teacher model results for the internal testing. I also felt that Figures 2 and 3 could be better explained, and that the y-axes could be adjusted so that the positive AND negative differences in mean performance could be read more easily. I also thought the numbers above each bar were redundant and could be incorporated into the dataset labels in parentheses.

---

### Decision · Program_Chairs · 2026-02-13

Accept (Poster)